# Novel *SCN5A* p.Val1667Asp Missense Variant Segregation and Characterization in a Family with Severe Brugada Syndrome and Multiple Sudden Deaths

**DOI:** 10.3390/ijms22094700

**Published:** 2021-04-29

**Authors:** Michelle M. Monasky, Emanuele Micaglio, Giuseppe Ciconte, Ilaria Rivolta, Valeria Borrelli, Andrea Ghiroldi, Sara D’Imperio, Anna Binda, Dario Melgari, Sara Benedetti, Predrag Mitrovic, Luigi Anastasia, Valerio Mecarocci, Žarko Ćalović, Giorgio Casari, Carlo Pappone

**Affiliations:** 1Arrhythmology Department, IRCCS Policlinico San Donato, San Donato Milanese, 20097 Milan, Italy; michelle.monasky@grupposandonato.it (M.M.M.); Emanuele.micaglio@grupposandonato.it (E.M.); g.ciconte@gmail.com (G.C.); valiborrelli91@gmail.com (V.B.); Sara.DImperio@grupposandonato.it (S.D.); Dario.Melgari@grupposandonato.it (D.M.); valerio.mecarocci@gmail.com (V.M.); zcalovic63@yahoo.com (Ž.Ć.); 2School of Medicine and Surgery, University of Milano-Bicocca, 20126 Monza, Italy; ilaria.rivolta@unimib.it (I.R.); anna.binda@unimib.it (A.B.); 3Laboratory of Stem Cells for Tissue Engineering, IRCCS Policlinico San Donato, San Donato Milanese, 20097 Milan, Italy; andrea.ghiroldi@gmail.com (A.G.); Anastasia.luigi@hsr.it (L.A.); 4Clinical Genomics—SMEL, IRCCS San Raffaele Hospital, 20132 Milan, Italy; benedetti.sara@hsr.it (S.B.); casari.giorgio@hsr.it (G.C.); 5Clinical Center of Serbia, Department of Emergency Cardiology, Cardiology Clinic, School of Medicine, University of Belgrade, 11000 Belgrade, Serbia; predragm@eunet.rs; 6Vita-Salute San Raffaele University, 20132 Milan, Italy

**Keywords:** Brugada syndrome, sudden cardiac death, genetic testing, *SCN5A*, risk stratification, variant, patch-clamp

## Abstract

Genetic testing in Brugada syndrome (BrS) is still not considered to be useful for clinical management of patients in the majority of cases, due to the current lack of understanding about the effect of specific variants. Additionally, family history of sudden death is generally not considered useful for arrhythmic risk stratification. We sought to demonstrate the usefulness of genetic testing and family history in diagnosis and risk stratification. The family history was collected for a proband who presented with a personal history of aborted cardiac arrest and in whom a novel variant in the *SCN5A* gene was found. Living family members underwent ajmaline testing, electrophysiological study, and genetic testing to determine genotype-phenotype segregation, if any. Patch-clamp experiments on transfected human embryonic kidney 293 cells enabled the functional characterization of the *SCN5A* novel variant *in vitro*. In this study, we provide crucial human data on the novel heterozygous variant NM_198056.2:c.5000T>A (p.Val1667Asp) in the *SCN5A* gene, and demonstrate its segregation with a severe form of BrS and multiple sudden deaths. Functional data revealed a loss of function of the protein affected by the variant. These results provide the first disease association with this variant and demonstrate the usefulness of genetic testing for diagnosis and risk stratification in certain patients. This study also demonstrates the usefulness of collecting the family history, which can assist in understanding the severity of the disease in certain situations and confirm the importance of the functional studies to distinguish between pathogenic mutations and harmless genetic variants.

## 1. Introduction

The genetics of Brugada syndrome (BrS) is currently a rapidly growing area of study [1]. It is associated with an increased risk for sudden cardiac death (SCD) due to ventricular tachycardia/fibrillation (VT/VF), many times in young, seemingly healthy individuals. The cardiac event rate in diagnosed patients over a median follow-up of 31.9 (14 to 54.4) months has been reported to be 5% overall, which increased to 7.7% in patients with aborted SCD, while family history of SCD or the presence of an *SCN5A* mutation were not predictive of arrhythmic events [2]. Despite this, only variants in the *SCN5A* gene are undisputedly recognized as causative of BrS, although over 20 genes are currently included in diagnostic genetic testing panels [3]. Nevertheless, the understanding of individual *SCN5A* variants is limited, as they can be classified as pathogenic, benign, or of uncertain significance, and even the variants known to be pathogenic are associated with a wide array of pathologies, including atrial standstill, atrial fibrillation, left ventricular non-compaction, dilated cardiomyopathy, long QT syndrome (LQTS), idiopathic ventricular fibrillation, and heart block [4], making the interpretation of novel variants difficult. Even worse, more than half of BrS patients do not harbor a variant in any of the genes in the BrS testing panel [1], leaving physicians and researchers with difficulties in finding the familial variant responsible for the syndrome and often without the possibility of using genetic testing to predict the disease presence. However, even for the patients in whom an *SCN5A* variant is found, the immediate clinical usefulness of detecting *SCN5A* variants that are considered of unknown significance (VUS) is under debate [5]. Thus, having conclusive data about particular variants would make genetic testing more useful and powerful in the clinic.

Since current knowledge about the genetics of BrS is generally of poor value in determining the care of individual patients, the diagnosis is performed either by the spontaneous presence of the BrS pattern on an electrocardiogram (ECG) or by invasive and risky pharmacological testing. A normal ECG cannot rule out the presence of BrS due to the transient nature of the characteristic ECG pattern, which can be modified by several factors, including changes in vagal tone or temperature [4]. Pharmacological tests have several limitations, including the need of specialized centers, the costs, and the time needed to wait for an appointment. On the other hand, genetic testing can be performed on saliva or blood samples, making collection from the patient easier and reaching a larger number of people. Thus, it would be extremely useful if it could predict disease presence and severity. 

In the present study, we report and functionally characterize for the first time the novel missense heterozygous variant NM_198056.2:c.5000T>A (p.V1667D) in the *SCN5A* gene and its segregation in a family with severe BrS and several sudden deaths. 

## 2. Material and Methods

### 2.1. Subjects

The patient and his relatives were evaluated in our Department of Arrhythmology and Electrophysiology at the hospital Policlinico San Donato. The study was conducted in accordance with the Declaration of Helsinki, and written informed consent of human subjects was obtained for their participation in the study and for publication. The procedures employed were reviewed and approved by the local Ethics Committee (approver number: M-EC-006/A, rev. 1 March 2013). 

### 2.2. DNA Extraction and Genetic Analysis

Genomic DNA was extracted from peripheral blood of the proband using the Maxwell 16 Blood DNA Purification kit (Promega). Quality and concentration was determined by Nanodrop (ThermoScientific) and Qbit (ThermoFisher). Samples were enriched using Tru Sight One Sequencing kit (Clinical exome, Illumina) and sequenced on NextSeq500 platform (Illumina). Sequences were analyzed according to GATK Best Practice criteria exploiting pipelines based on BWA, Smith-Waterman Algorithm, freebayes, SnpSift-SnpEFF and BaseSpace Onsite. The Next Generation Sequencing (NGS) panel contained 50 genes described in BrS research literature (*ABCC9, ACTC1, AKAP9, ANK2, CACNA1C, CACNA2D1, CACNB2, CALM1, CALM3, CAV3, DSC2, DSG2, DSP, FLNC, GPD1L, HCN4, JUP, KCND2*, *KCND3*, *KCNE1*, *KCNE1L*, *KCNE2*, *KCNE3*, *KCNE5*, *KCNH2*, *KCNJ2*, *KCNJ5*, *KCNJ8*, *KCNQ1*, *MOG1/RANGRF*, *MYBPC3*, *MYH7*, *MYL2*, *MYL3*, *PKP2*, *PLN*, *RYR2*, *SCN10A*, *SCN1B*, *SCN2B*, *SCN3B*, *SCN4B*, *SCN5A*, *SEMA3A*, *SNTA1*, *TNNI3*, *TNNT2*, *TPM1*, *TRDN*, *TRPM4*). The mean coverage of the sequenced regions was 134X. The analyzed regions covering < 20X include the genes *AKAP9* (91709443–91709466), *SCN1B* (35521715–35521764) and *JUP* (39919235–39919244).

### 2.3. Plasmid Generation

Wild-Type (WT) *SCN5A* complementary DNA (cDNA) was subcloned into the pcDNA3.1 plasmid, with tandem twin Strep-tag and FLAG tag at the N terminus of Na_V_1.5. The mutation was engineered by site directed mutagenesis through the QuikChange II site-directed mutagenesis kit (Agilent Technologies), according to the manufacturer’s instructions. The oligonucleotides used were: forward, *5′*-CTGCTGCTCTTCCTCGACATGTTCATCTAC-*3′*, reverse, *5′*- GTAGATGAACATGTCGAGGAAGAGCAGCAG-*3′.* The construct was sequenced by Sanger technique by our group at San Raffaele Hospital to verify the correct introduction of the mutation and ensure the validity of the sequence. *SCN1B*, the gene encoding for the human cardiac β1 subunit, was subcloned in a pIRES vector engineered with EGFP that served as a reporter gene.

### 2.4. Cell Culture and Transfection

HEK293 (human embryonic kidney 293) cells were cultured in a controlled environment (5% CO_2_, 37 °C) and maintained in an appropriate medium (DMEM/F12; Euroclone) supplemented with 10% FBS, 2 mM L-Glutammine, 100 U/mL, and 100 μg/mL Pen/Strep. Equal amounts of *SCN5A* (0.5 µg, WT or p.V1667D) and *SCN1B* subunits cDNA (1:1 ratio) were transiently transfected using jetPRIME reagent (PolyPlus transfection, Euroclone) according to the manufacturer’s instructions. In order to mimic the genetic balance of affected individuals, 0.25 µg of the WT and 0.25 µg of the variant alpha subunit encoding plasmid, as well as 0.5 µg of beta 1 subunit, were co-transfected (0.5:0.5:1 ratio).

### 2.5. Functional Analysis

Whole-cell patch clamp experiments were performed at room temperature (RT) 48 h after transfection and using pipettes pulled to a 2–5 MΩ resistance (Model P-97 Sutter Instruments). Series resistance (Rs) were compensated and the compensation was readjusted before each voltage-clamp protocol. Briefly, cell medium was replaced with an extracellular solution composed of (mmol/L) 130 NaCl, 2 CaCl_2_, 5 CsCl, 1.2 MgCl_2_, 10 HEPES, 5 glucose, pH 7.4 adjusted with CsOH, while electrodes were filled with an intracellular solution containing (mmol/L) 135 CsCl, 5 NaCl, 5 KCl, 5 EGTA-KOH, 10 HEPES, 1 MgCl_2_, pH 7.2 adjusted with CsOH. Liquid Junction potential (LJP) was 13.96 mV. External sodium concentration was reduced to 40 mM using N-Methyl-D-glucamine as Na^+^ substitute when the voltage dependence of activation was investigated. LJP in this case was 8.23 mV. 

Holding potential was −100 mV. Steady-state activation was studied pulsing from −80 to +75 mV (+5 mV increment, 25 ms duration) while steady-state inactivation protocol had a first 500 ms duration step (from −140 to +10 mV, increment +10 mV) followed by a −10 mV test pulse (20 ms durations). Recovery from inactivation was studied through a protocol in which two −10 mV steps (500 ms and 20 ms duration respectively) were divided by a −100 mV pulse of increasing duration from 0.01 ms to 3 s. The time dependence of the onset of intermediate and slow inactivation was measured using a two pulse protocol: the first pulse P1 stepped from the holding potential of −100 mV to −10 mV (increasing duration from 1 to 1000 ms), then a step back to −100 mV for 20 ms let the channels recover from fast inactivation and finally the second pulse P2 to −10 mV (duration 20 ms) enables the recording of sodium current. The resulting P2/P1 ratio was normalized and plotted against the P1 duration [6,7]. 

Steady-state activation and availability curves were fitted with a Boltzmann function y = 1/(1 + exp((V − V_1/2_)/k)) where y is the relative current, V is the membrane potential, V_1/2_ is the half-maximal voltage and k is the slope factor. The onset of the fast inactivation process was studied fitting the decay of the peak current traces elicited by a 25 ms depolarizing pulse from −100 to +20 mV with a mono exponential function. Data were acquired with a Multiclamp 700B amplifier and Digidata 1440A (Axon Instruments, Molecular Device) and pClamp 10.3 software (Molecular Devices) and analyzed with Clampfit 10.3 software (Molecular Devices).

### 2.6. Statistical Analysis

Functional data are presented corrected for the LJP and as mean ± SD of at least three independent experiments; *n* denotes the number of cells. A two-way ANOVA was performed for multiple comparison, followed by a modified *t* test with Fisher correction (OriginPro 8; OriginLab). Values of *p* < 0.05 were considered significant and indicated with * in figures.

## 3. Results

### 3.1. Proband Patient Characteristics and Family History

The proband is a 48-year-old man with a family history of sudden death (Figure 1). His paternal grandfather’s brother died suddenly during the night at 40 years old. The proband’s paternal grandfather suddenly died during sleep at 69 years old. The proband’s father had suffered from recurrent syncope, which led to a pacemaker implantation elsewhere at 50 years old, but the syncopal episodes continued even after pacemaker implantation. Proband’s father then died suddenly during sleep at 69 years old without further information available. The proband suffered from objective vertigo and dizziness from 44 years old. For this reason, he performed an ear, nose, and throat (ENT) evaluation, an audiometric exam, and a brain MRI, all with unremarkable results. At the age of 47 years old, the proband experienced a cardiac arrest while watching a sports match. He was resuscitated, and an unspecific BrS pattern was identified on the ECG. Therefore, he came to our center, and a spontaneous type 1 BrS ECG pattern was observed, which confirmed the BrS diagnosis (Figure 2). An ICD was implanted. 

### 3.2. Genetic Testing Results and In Silico Prediction

The genetic testing performed on the patient revealed the novel heterozygous missense variant NM_198056.2:c.5000T>A (p.V1667D) in the *SCN5A* gene (LOVD: https://databases.lovd.nl/shared/individuals/00314832; accessed on 28 April 2021), which was confirmed by Sanger sequencing (Figure 3). No other variant was detected in any of the genes screened. 

The c.5000T>A variant was classified as likely pathogenic according to ACMG criteria [8,9]:PM1, Moderate: UniProt protein SCN5A_HUMAN trans-membrane region ‘Helical’ has 4 non-VUS missense/in-frame/non-synonymous, variants (4 pathogenic and 0 benign), pathogenicity = 100.0% which is more than threshold 50.0%.PM2, Moderate: Variant not found in gnomAD exomes (good gnomAD exomes coverage = 99.6). Variant not found in gnomAD genomes (good gnomAD genomes coverage = 31.9).PM5, Moderate: A nearby variant chr3:38592864 G>A (Val1649Ile) is classified Pathogenic by UniProt Variants (and confirmed using ACMG).PP2, Supporting: The gnomAD missense Z-Score = 2.75 is greater than 0.647.PP3, Supporting: Pathogenic computational verdict based on 11 pathogenic predictions from BayesDel_addAF, DEOGEN2, EIGEN, FATHMM-MKL, LIST-S2, M-CAP, MVP, MutationAssessor, MutationTaster, PrimateAI, and SIFT vs. no benign predictions (1 uncertain prediction from DANN).

### 3.3. Assessment of Family Members

Due to the family history, proband’s sisters and children were evaluated. The proband’s 42-year-old and 37-year-old sisters are both asymptomatic, and their only remarkable condition upon presentation at our center was autoimmune thyroiditis that was discovered from a routine blood test. Due to family history, they underwent ajmaline challenge at our facility, which resulted negative for the older sister, but positive for the younger (Figure 4). Genetic testing by Sanger sequencing was positive in the younger sister for the familial variant NM_198056.2:c.5000T>A (p.V1667D) in the *SCN5A* gene. The older sister reported to us by telephone that her genetic test for the same variant resulted negative.

The proband’s children, a 15-year-old son and a 13-year-old daughter, are asymptomatic. They both underwent ajmaline challenge at our facility, which resulted negative (Figure 4). Additionally, their genetic testing by Sanger sequencing was negative for the familial variant NM_198056.2:c.5000T>A (p.V1667D) in the *SCN5A* gene.

The proband’s 67-year-old mother was reported by the proband to have tested negative elsewhere for the familial variant in the *SCN5A* gene. She has not undergone an ajmaline test. 

### 3.4. Functional Analysis of p.V1667D Na_V_1.5 Channels

The p.V1667D mutant channel was transiently overexpressed in HEK293 cells in the presence of the beta1 subunit. Its functional properties were investigated by patch-clamp experiments in whole cell configuration. In order to mimic the genetic background of the patients, the mutated channel was also co-expressed with the Nav1.5 WT. Figure 5 and Table 1 summarize the results found and compare them with the ones obtained from the WT protein. The level of the expression was highly variable, especially for the WT channel; nevertheless, the presence of the mutation significantly reduced the magnitude of the current density and delayed the kinetics of both the activation and fast inactivation, and the time course of recovery from the inactivation (the time to peak, τ and Recovery from inactivation t2, respectively in Table 1). The voltage dependence of the activation was not affected, while the availability curve was 5 mV right-shifted in cells expressing only the mutated form of the channel. Finally, we did not observe mutation-related differences in the development of the intermediate and slow inactivation process.

## 4. Discussion

In the present study, we report for the first time the heterozygous variant NM_198056.2:c.5000T>A (p.V1667D) in the *SCN5A* gene, both generally and in BrS. In addition to demonstrating the disease association for this variant, we reveal herein the severity of the phenotype associated with this mutation, as three members of the same family have died suddenly at night or while sleeping, and a fourth has had aborted cardiac arrest. In a study evaluating the long-term prognosis of patients diagnosed with BrS, during a median follow-up of 31.9 months, 5% of patients experienced a cardiac event (appropriate ICD shock or sudden death) [2]. In the present study, instead, three family members suddenly died, and, of the two living family members who carry the mutation, the proband has suffered an aborted cardiac arrest, while his sister is still younger than the age at which any of her family members first experienced a cardiac event. At this point, it is not possible to rule out the possibility of her having a cardiac arrest. Thus, it appears that the rate of mortality in this family is unusually high. 

Regarding the arrhythmic risk stratification, large registries do not associate a family history of SCD or the presence of variants in the *SCN5A* gene with an increased risk of ventricular fibrillation [10]. However, in the present study, it is evident that data collected from family members can be extremely useful to the physician to understand the severity of the phenotype resulting from a genetic variant and in determining the clinical management. When a family with a strong history of sudden death, such the one described herein, is presented to the clinic, it offers a great opportunity for the understanding of the association of a particular genetic variant with the severity of the disease. In this case, the genetic testing could be used to suggest family members in need of an ICD, even if current guidelines do not rely on genetic testing for risk stratification. *In silico* studies discussed above [9], as well as the cellular electrophysiological data obtained in HEK293 cells, support the hypothesis of a severe pathogenic effect of the p.V1667D variant. Human induced pluripotent stem cells (iPSCs) are an important model because they carry specific cardiomyopathy-associated mutations and allow for personalized medicine [11]. However, HEK293 cells are also a useful model because of their development of recombinant proteins or adenovirus productions, their human-like posttranslational modification of protein molecules, their high transfection efficiency yielding high-quality recombinant proteins, and their easy maintenance and expression of high fidelity membrane proteins, such as ion channels and transporters [12]. The p.V1667D variant induced a loss of function in the Na_V_1.5 channel, which displayed a reduced magnitude of the current density of about 40% and an impairment of the kinetics of the activation and recovery from inactivation when mimicking the patients’ genetic substrate in HEK293 cells. This strongly suggests that the radical shift of amino-acid class from a hydrophobic residue (V) to a polar, negatively charged one (D), is the bona fide cause of the proband’s phenotype in the current study. To be thorough, patch-clamp data obtained from cells expressing the p.V1667D variant alone, thus mimicking a homozygous hypothetical patient, showed a right-shift in the availability curve, suggesting in this case a higher number of channels ready to be primed following a stimulus, a feature more typical of a gain of function as well as the slowing in the inactivation kinetics. We report this observation for completeness of information, pointing out, though, that all the carriers here described are heterozygous for the *SCN5A* variant.

Despite the recent literature discussing whether BrS is a far more complex disorder [13,14], it has been historically believed to be a Mendelian disease, caused by a single mutation in a single gene. Variants detected in BrS patients have more often been described in *SCN5A* than in any other gene [4] and for this reason, some of them have been described as a monogenic cause of BrS. Moreover, the causative role for every gene in the BrS genetic testing panels has been disputed, with the exception of *SCN5A* [3]. 

The *SCN5A* gene encodes the sodium voltage-gated channel alpha subunit 5 (Na_V_1.5) protein [4]. This protein is essential for the fast influx of sodium ions across the cell membrane that results in the upstroke phase of the action potential. The proper functioning of this protein is crucial for the systolic increase in sodium during the action potential that drives the movement of other cations in a delicate balance. Thus, the improper functioning of this channel may cause fatal arrhythmias [4].

The nucleotide c.5000T is located in exon 28 of the *SCN5A* gene and the frequency of the c.5000T>A variant in the general population is 1/251,492 [9]. According to Varsome [9], no variants of unknown significance are reported in the corresponding amino acidic residue 1667, in which valine is conserved throughout evolution (GERP score 4.6799) and among voltage-dependent sodium channels of different tissues and different species [15]. Interestingly, it is located in a transmembrane spanning region, as happens in the 74% of BrS patients hosting a single *SCN5A* mutation [1]. In particular, it is part of the S5 in the fourth domain, a segment that may play a role in activation-inactivation coupling [16]. Thus, it may not be surprising that also the p.V1667D variant induced an impairment in the availability curve. 

Another mutation in the same residue, p.V1667I, was previously reported in a large Finnish family in which two of the nine carriers were symptomatic with a mild LQTS that worsened after the administration of drugs such as epinephrine and halofantrine [15,16]. Moreover, the same mutation was associated with a BrS phenotype in a single patient of the AMC Heart Center in Amsterdam [1]. The overexpression of the p.V1667I variant in the absence of the wildtype channel in tsa-201 cells resulted overall in a gain of function due to multiple mechanisms [16], consistent with a LQTS clinical phenotype. 

The predictive value of *SCN5A* mutations for BrS risk remains a hotly debated topic [1,17,18,19]. While many studies have attempted to understand the relationship between *SCN5A* variants and various clinical parameters, these studies seem to have grouped together any and all *SCN5A* variants, regardless of the *in silico* predictions of clinical significance. Despite grouping together all variants within *SCN5A*, regardless of predicted significance, these studies found that the presence of any *SCN5A* variant was, on average, associated with more conduction abnormalities detectable on the ECG [17,18], including a higher rate of detecting a spontaneous type-1 BrS ECG pattern, more pronounced conduction or repolarization abnormalities, increased atrial vulnerability [17], a higher positive rate of late potentials, and longer P-wave, PQ, and QRS durations [18]. Studies have also reported that BrS patients with an *SCN5A* variant have a higher risk for cardiac events [18], including a younger age at the onset of symptoms [17,18], and an overall worse prognosis [17]. A study with multivariate analysis reported that significant predictors of cardiac events were limited to only history of aborted cardiac arrest and the presence of an *SCN5A* mutation [18]. In one study, the presence of any *SCN5A* mutation resulted in 32% sensitivity and 57% specificity in identifying patients with cardiac arrest [19]. However, again, these studies did not distinguish between *SCN5A* mutations predicted as pathogenic (or predicted non-disease causing) and VUS. 

Given the fiercely disputed issue of the predictive value of *SCN5A* variants, they are currently regarded by many as not being useful for the prediction of future cardiac arrhythmic events [19]. Guidelines for the management of BrS state that genetic test results should be used to determine only disease probability, rather than being “binary/deterministic in nature” [20,21,22]. However, our data clearly demonstrates the importance of genetic testing and the usefulness of identifying the *SCN5A* variants that occur with disease segregation, enabling the prediction of which family members carry the disease and even assisting in risk stratification. 

## 5. Conclusions

The novel missense heterozygous variant NM_198056.2:c.5000T>A (p.V1667D) in the *SCN5A* gene causes a loss of function of the protein and segregates with a severe BrS phenotype and sudden cardiac death in the family presented, demonstrating the usefulness of genetic testing for diagnosis and risk stratification in certain patients. This data also demonstrates the usefulness of both collecting the family history and performing functional studies, which can both assist the clinician in understanding the severity of the disease in certain situations.

## Figures and Tables

**Figure 1 ijms-22-04700-f001:**
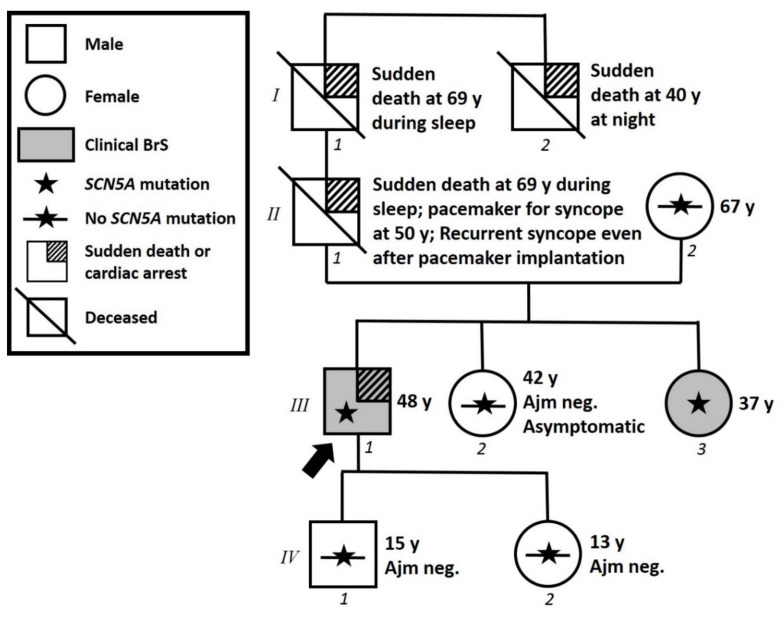
**Family pedigree.** Proband identified with arrow. Square: male; Circle: female; Shaded: clinically affected by Brugada syndrome; Star: molecularly confirmed *SCN5A* variant; Star with slash: negative for *SCN5A* variant; y = years old at diagnosis.

**Figure 2 ijms-22-04700-f002:**
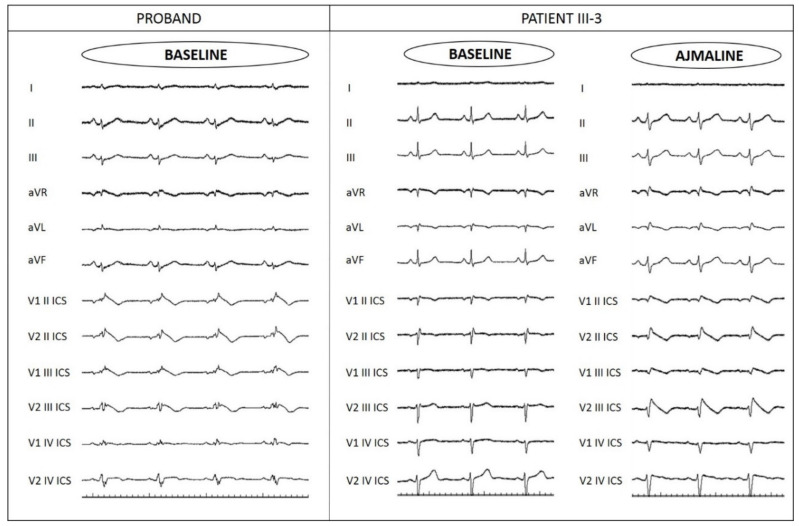
Left panel shows proband’s ECG with a spontaneous type 1 BrS ECG pattern. Right panels show patient III-3, one of the proband’s sisters, with an ajmaline-induced type 1 BrS ECG pattern.

**Figure 3 ijms-22-04700-f003:**
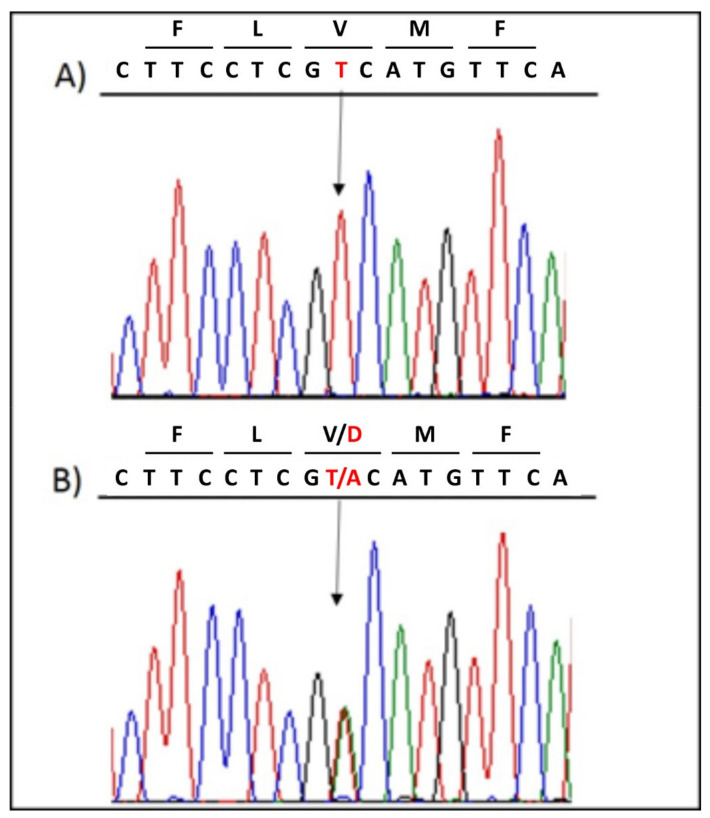
**Post-NGS Sanger confirmation of the presence of the heterozygous variant *SCN5A* c.5000 T>A.** (**A**) Direct sequencing electropherograms show the wild-type sequence in a family member. (**B**) The presence of the single nucleotide variant in the proband is denoted by an arrow. The first row of letters in each panel indicate the amino acid that is specified by the codon below.

**Figure 4 ijms-22-04700-f004:**
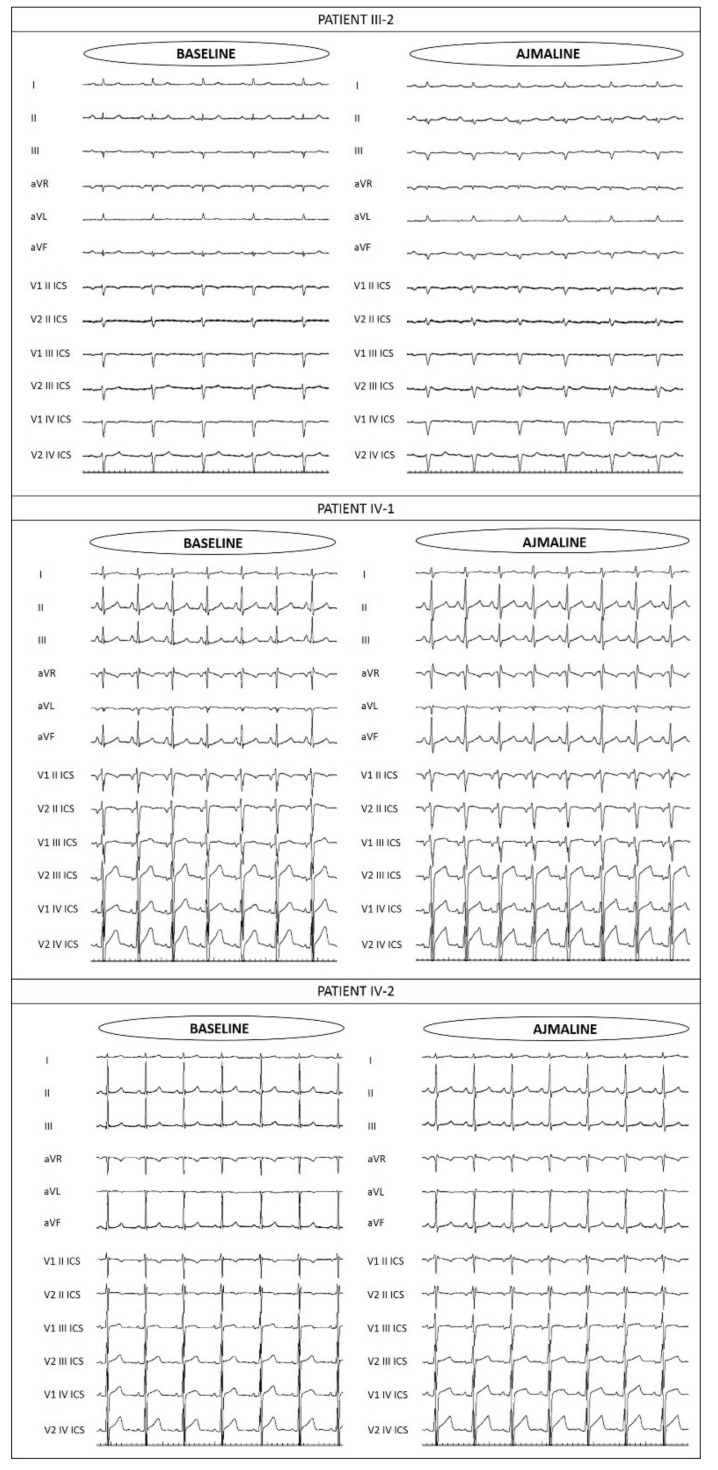
Electrocardiograms before and after ajmaline administration, demonstrating that these family members were negative for BrS, due to the lack of the development of a type 1 BrS ECG pattern.

**Figure 5 ijms-22-04700-f005:**
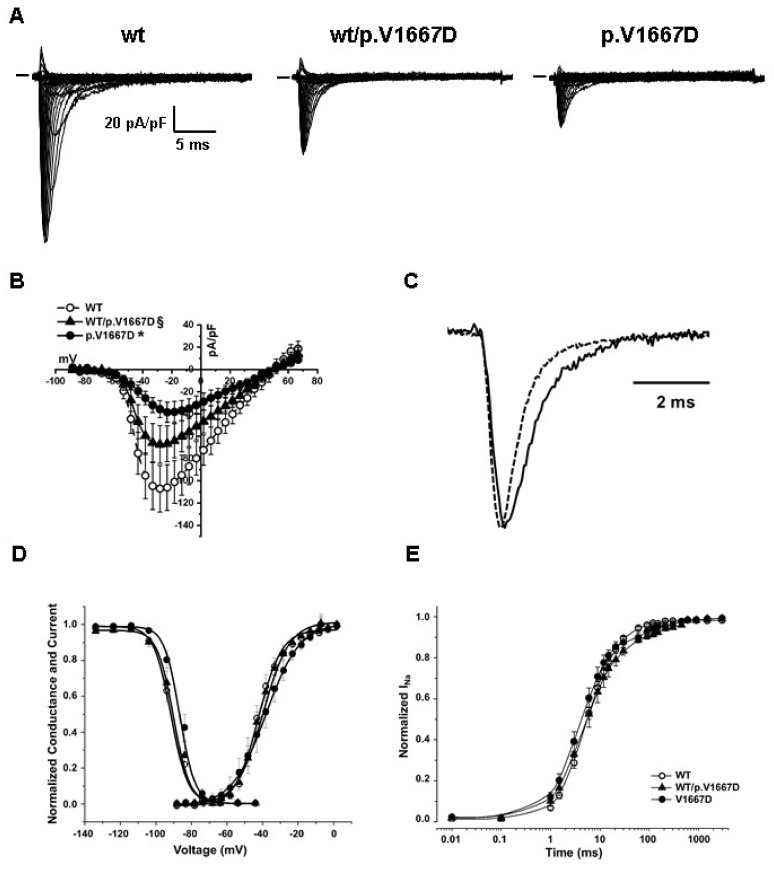
**Functional characterization of the p.V1667D variant in HEK293 cells.** (**A**) Typical families of current traces recorded from HEK cells expressing WT, WT/p.V1667D, and p.V1667D Nav1.5 channel. The current amplitude measured in 40 mM extracellular Na^+^ was normalized by the cell capacitance. (**B**) Averaged current-voltage relationship (empty circles for WT, filled circles for p.V1667D and triangles for WT/p.V1667D). (**C**) Comparison of normalized WT (dashed line) or p.V1667D (solid line) Nav1.5 current elicited by a depolarizing step @-20 mV showing the impairment in the kinetics of the activation and fast inactivation process. (**D**) Steady state activation and availability curve. (**E**) Recovery from the inactivation. Statistics and numbers of cells tested are reported in Table 1.

**Table 1 ijms-22-04700-t001:** Functional characterization of the p.V1667D variant in HEK293 cells and its effect on the wildtype current. (The number in parenthesis represents the cells tested and considered for the analysis; * *p* < 0.05 vs. WT).

		WT	WT/p.V1667D	p.V1667D
Current density @-28 mV (pA/pF)		−107.11 ± 21 (*n* = 16)	−67.30 ± 17 * (*n* = 16)	−35.58 ± 7 * (*n* = 13)
Time to peak (@-28 mV, msec)		0.58 ± 0.02	0.70 ± 0.03 *	0.69 ± 0.04*
Kinetic of fast inactivation (@-28 mV)	τ (ms)	0.75 ± 0.02 (*n* = 17)	1.02 ± 0.1 * (*n* = 10)	0.94 ± 0.04 * (*n* = 10)
Steady state of activation	V_1/2_ (mV)	−42.6 ± 1.8 (*n* = 16)	−41.6 ± 1.5 (*n* = 16)	−38.9 ± 3.5 (*n*=13)
	k	4.5 ± 0.3	4.8 ± 0.1	5.3 ± 0.5 *
Availability curve	V_1/2_ (mV)	−90.9 ± 1.3 (*n* = 20)	−89.3 ± 1.9 (*n* = 13)	−85.8 ± 1.6 * (*n* = 12)
	k	4.8 ± 0.2	4.8 ± 0.3	4.6 ± 0.3
Recovery from inactivation	t1 (ms)	7.0 ± 0.5 (*n* = 18)	6.0 ± 1 (*n* = 8)	5.4 ± 0.5 (*n* = 11)
	t2 (ms)	67.30 ± 14	193.3 ± 31 *	142.5 ± 22 *
Development of intermediate inactivation	t1 (ms)	485 ± 114 (*n* = 9)	492 ± 253 (*n* = 9)	469 ± 178 (*n* = 6)

## Data Availability

Not applicable.

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
