# Peer review of "Novel SCN5A p.Val1667Asp Missense Variant Segregation and Characterization in a Family with Severe Brugada Syndrome and Multiple Sudden Deaths"

_ijms, 2021, doi:10.3390/ijms22094700_

Round 1

Reviewer 1 Report

Monasky et al.  presented a very well documented family case with severe Brugada syndrome (BrS). In the family with frequent sudden cardiac  death  Authors identified a novel p.Val1667Asp variant in the SCN5A gene. The pathogenicity of the variant is strentghtened by the fact that another aa substitution in the same position of the polypeptide chain is associated with arrhythmia in Finnish family, and the same mutation was associated with BrS phenotype in a sporadic case.

The description is clear, both clinical  and experimental data on the variant are nicely presented.

Two questions:

From the description of the family we can learn that 2 sudden deaths occurred rather late in the victims’ lives, namely  at 69 years, one was at 40y, and sudden cardiac arrest at age of 48 years.

It is well known that co-existence of arrhythmic predisposition and ischemia may exacerbate the course of CAD and increase the risk of sudden cardiac death. What about coronary angiography, or coronary CT scan in the family?

Second issue relates to genotype-phenotype correlations. Authors  used  a panel of genes containing 50 genes described in BrS. Were there any additional variants e.g. in SCN10A in the proband?

Author Response

Thank you for your time, your comments, and your positive review. We hope that you find our responses satisfactory and that our manuscript is now suitable for publication. Thank you again.

Comment: From the description of the family we can learn that 2 sudden deaths occurred rather late in the victims’ lives, namely  at 69 years, one was at 40y, and sudden cardiac arrest at age of 48 years.

It is well known that co-existence of arrhythmic predisposition and ischemia may exacerbate the course of CAD and increase the risk of sudden cardiac death. What about coronary angiography, or coronary CT scan in the family?

Response: The arrhythmic phenotype exhibited in the proband’s family has been characterized by normal serum cholesterol without any clue suggesting ischemic cardiomyopathy. Therefore, no coronary angiography nor coronary TC scan were performed.

Comment: Second issue relates to genotype-phenotype correlations. Authors  used  a panel of genes containing 50 genes described in BrS. Were there any additional variants e.g. in SCN10A in the proband?

Response: No. We now write, “No other variant was detected in any of the genes screened.”

Reviewer 2 Report

In the manuscript 'Novel SCN5A p.Val1667Asp Missense Variant Segregation and Characterization in a Family with Severe Brugada Syndrome and Multiple Sudden Deaths, submitted by Monasky et al. the authors identified a novel SCN5A missense variant in a family with Bruguada syndrome and characterized this mutation via patch clamp measurements. The manuscript is of general interest and worth to publish. Functional data on specific mutations are frequently missing and the authors are correct that such studies help also for classification and interpretation of genetic variants. However, the manuscripts need several minor changes:

1.) Please indicate also in the abstract that the patch clamp measurements were performed in vitro using transfected HEK293 cells.

2.) Please mention in the introduction that mutations in cardiomyopathy (e.g. ARVC) associated genes like DES could also cause sudden cardiac death. For example the mutation DES-p.A120D causes sudden cardiac death and cite relevant literature (see Brodehl A et al. 2013 Dec;6(6):615-23. doi: 10.1161/CIRCGENETICS.113.000103. Epub 2013 Nov 7. The novel desmin mutant p.A120D impairs filament formation, prevents intercalated disk localization, and causes sudden cardiac death.).

3.) Please write the gene name SCN5A always in Italics.

4.) Please list all (!) genes screened by NGS in an Appendix and remove the gene names from paragraph 2.2.

5.) Please indicate the kit used for DNA extraction and describe library preparation and sequencing in more detail.

6.) Please indicate the oligonucleotides used for site directed mutagenesis.

7.) Please indicate were the Sanger sequencing has been done.

8.) Please list also the amino acid sequnece above the nucleotide sequence in Figure 3.

9.) Please discuss also the relevance of HEK293 cells for an electrophysiological study. Many studies use cardiomyocytes derived from induced pluripotent stem cells. Please cite relevant literature (e.g. Human Induced Pluripotent Stem-Cell-Derived Cardiomyocytes as Models for Genetic Cardiomyopathies. Brodehl A, Ebbinghaus H, Deutsch MA, Gummert J, Gärtner A, Ratnavadivel S, Milting H. Int J Mol Sci. 2019 Sep 6;20(18):4381. doi: 10.3390/ijms20184381.).

  10.) Please update the Funding section and Data Availablility Statement.   Good luck with the revision!  

Author Response

Thank you for your time, your comments, and your positive review. We hope that you find our responses satisfactory and that our manuscript is now suitable for publication. We list our responses below point-by-point. Thank you again.

1.) Please indicate also in the abstract that the patch clamp measurements were performed in vitro using transfected HEK293 cells.

Done!

2.) Please mention in the introduction that mutations in cardiomyopathy (e.g. ARVC) associated genes like DES could also cause sudden cardiac death. For example the mutation DES-p.A120D causes sudden cardiac death and cite relevant literature (see Brodehl A et al. 2013 Dec;6(6):615-23. doi: 10.1161/CIRCGENETICS.113.000103. Epub 2013 Nov 7. The novel desmin mutant p.A120D impairs filament formation, prevents intercalated disk localization, and causes sudden cardiac death.).

We fully agree that mutations in DES and other genes might be responsible for sudden cardiac death in general. In this paper, we aimed to discuss this one particular gene in Brugada syndrome. The patient did not display signs of ARVC, and thus this aspect was not discussed because we felt that it is outside the scope of the current study.

3.) Please write the gene name SCN5A always in Italics.

We had written it always in Italics in the submitted document. However, the journal typeset it without the Italics. We will have to check this on the proof carefully!

4.) Please list all (!) genes screened by NGS in an Appendix and remove the gene names from paragraph 2.2.

Since which genes were screened is a common question among researchers, we think that it would be most convenient for most readers if the information was located in the main document, rather than in an appendix, which we fear could get put as supplementary information by the journal and be downloadable as a separate document. Thus, if it’s okay, we would prefer to leave it in the main document.

5.) Please indicate the kit used for DNA extraction and describe library preparation and sequencing in more detail.

We now write, “Genomic DNA was extracted from peripheral blood of the proband using the Maxwell 16 Blood DNA Purification kit (Promega). Quality and concentration was determined by Nanodrop (ThermoScientific) and Qbit (ThermoFisher). Samples were enriched using Tru Sight One Sequencing kit (Clinical exome, Illumina) and sequenced on NextSeq500 platform (Illumina). Sequences were analyzed according to GATK Best Practice criteria exploiting pipelines based on BWA, Smith-Waterman Algorithm, freebayes, SnpSift-SnpEFF and BaseSpace Onsite.”

6.) Please indicate the oligonucleotides used for site directed mutagenesis.

The sequences of the oligo used for the mutagenesis have been inserted in the Materials and Methods section.

7.) Please indicate were the Sanger sequencing has been done.

We now write, “The construct was sequenced by Sanger technique by our group at San Raffaele Hospital to verify the correct introduction of the mutation and ensure the validity of the sequence.”

8.) Please list also the amino acid sequence above the nucleotide sequence in Figure 3.

Done.

9.) Please discuss also the relevance of HEK293 cells for an electrophysiological study. Many studies use cardiomyocytes derived from induced pluripotent stem cells. Please cite relevant literature (e.g. Human Induced Pluripotent Stem-Cell-Derived Cardiomyocytes as Models for Genetic Cardiomyopathies. Brodehl A, Ebbinghaus H, Deutsch MA, Gummert J, Gärtner A, Ratnavadivel S, Milting H. Int J Mol Sci. 2019 Sep 6;20(18):4381. doi: 10.3390/ijms20184381.).

We now write, “Human induced pluripotent stem cells (iPSCs) are an important model because they carry specific cardiomyopathy-associated mutations and allow for personalized medicine[11]. However, HEK293 cells are also a useful model because of their development of recombinant proteins or adenovirus productions, their human-like posttranslational modification of protein molecules, their high transfection efficiency yielding high-quality recombinant proteins, and their easy maintenance and expression of high fidelity membrane proteins, such as ion channels and transporters[12].”

  10.) Please update the Funding section and Data Availablility Statement.   Good luck with the revision! 

This was again written by the typesetter! We will check carefully the proof. Thank you!